# Bootstrapping Zero-Shot Reasoning in Small Language Models via Advantage-Weighted Self-Distillation

## Abstract

Small language models (0.5B–3B) typically lack mathematical reasoning ability, often scoring near 0% on tasks they can solve with few-shot demonstrations. Existing approaches rely on thousands of supervised chain-of-thought (CoT) traces or complex multi-round self-distillation pipelines. We introduce *Advantage-Weighted Direct Preference Optimization* (AWDPO), a lightweight alignment method that bridges the gap between few-shot and zero-shot reasoning. Unlike prior approaches, AWDPO formulates training as a single-pass preference optimization objective that aligns a model's zero-shot distribution with its own few-shot behavior. Our loss combines an advantage-weighted preference term with a dynamic MLE anchor, yielding stable training and implicit trust-region regularization.

On GSM8K, AWDPO transforms Qwen-2.5 base models (0.5B–3B) from 0% to 39%–77% accuracy, recovering over 90% of a supervised fine-tune that uses 7,473 CoT traces — a 1,750× reduction in CoT data. The method generalizes to SVAMP, ASDiv, and MATH-500, where AWDPO recovers up to 90% of supervised CoT performance. Our analysis shows that AWDPO is equivalent to a Kullback-Leibler (KL)-constrained policy improvement step under projected DPO. These results demonstrate that small base models can substantially improve their mathematical reasoning ability from minimal supervision, providing a principled and data-efficient alternative to supervised CoT or Reinforcement Learning (RL)-based methods for mathematical reasoning.

## 1 Introduction

Mathematical problem-solving is a gold-standard benchmark for evaluating reasoning in large language models (LLMs) (Cobbe et al., 2021). Large instruction-tuned models such as GPT-4 (OpenAI, 2023) achieve impressive accuracy on challenging math benchmarks like GSM8K. Recent work has shown that reinforcement learning (RL) can further enhance these reasoning capabilities, with models like DeepSeek-R1 (DeepSeek-AI et al., 2025) demonstrating substantial improvements through RL-based post-training.

Separately, research on chain-of-thought prompting (Wei et al., 2022) has revealed that providing step-by-step demonstrations can dramatically improve reasoning performance in LLM's. This suggests that reasoning abilities may be latent within these models, requiring only the right contextual scaffolding to emerge.

Our work lies at the intersection of these two paradigms. We propose **Advantage-Weighted Direct Preference Optimization (AWDPO)**, an optimization objective that aligns a model's zero-shot distribution with its own few-shot behavior. Unlike iterative trace-filtering methods, AWDPO directly modifies the preference loss: we weight updates by the advantage between few-shot and zero-shot responses, thereby learning from both successes and failures. To stabilize training under parameter-efficient adaptation, we add a lightweight MLE anchor on correct exemplars, yielding an implicit trust region similar to KL-regularized RL methods.

This approach addresses a critical gap: while large models benefit from extensive post-training and small models can leverage few-shot prompting, there has been limited work on efficiently transferring reasoning capabilities into the weights of compact, base language models without massive supervision.

Our method requires neither a large scale external teacher nor reinforcement learning. Instead, we leverage the base model's chain-of-thought output in a few-shot setting as a pseudo-teacher for the same model in a zero-shot setting. We compute a direct preference loss (Rafailov et al., 2023) between the model's few-shot and zero-shot outputs, weighting the update by the *advantage*, or the difference in a task reward between the two outputs. Concurrently, we anchor the model on a set of the model's own completely correct solutions using a dynamically scaled maximum-likelihood estimation (MLE) loss. By combining these signals, AWDPO internalizes the benefits of few-shot chain-of-thought reasoning into the model's parameters, so that the model can perform reasoning without context exemplars at inference time.

Crucially, AWDPO avoids the pitfalls of unconstrained preference optimization by enforcing two implicit *trust regions*. First, we maintain *model fidelity* to the original model by applying low-rank adaptation (LoRA; Hu et al., 2022) and assuming a Lipschitz-bounded policy change, which keeps parameter updates small and implicitly serves a role similar to a Kullback-Leibler (KL)-regularization in RL-based methods. Second, we preserve *teacher fidelity* via an MLE anchor on correct few-shot demonstrations, preventing the model from diverging or collapsing when the pseudo-teacher's quality is imperfect. We introduce an *online loss-balancing* mechanism that scales the MLE term to match the preference loss magnitude, eliminating the need for manual weight tuning between objectives.

We evaluate AWDPO on challenging math word problem benchmarks. On GSM8K (Cobbe et al., 2021), using up to 4 chain-of-thought exemplars and 7k answer only QA pairs, AWDPO attains over 90% of the accuracy gains of a fully-supervised 7k-example fine-tuning, despite using $1/1750$-th of the high quality chain-of-thought data. It outperforms the base model's zero-shot and few-shot performance, as well as baselines using vanilla DPO (no advantage weighting) and an unweighted preference loss. AWDPO-trained 0.5B and 1.5B models reach *near-parity*, demonstrating that targeted self-distillation can sometimes outperform broad supervised exposure. Moreover, the AWDPO-trained models generalize their reasoning skills to other datasets (SVAMP (Patel et al., 2021), ASDiv (Miao et al., 2020), and MATH-500 (Lightman et al., 2024; Hendrycks et al., 2021)) in a zero-shot setting, achieving a large fraction of the fully-supervised performance on those benchmarks as well.

In summary, our contributions are:

1. We propose AWDPO, a data-efficient optimization method that internalizes few-shot reasoning into compact models' zero-shot behavior, requiring only a tiny fixed set of exemplars with answer-key rewards, and avoiding both large external teachers and reinforcement learning.
2. We show how using low-rank adapter fine-tuning and a Lipschitz constraint implicitly bounds policy drift (providing a KL-like regularization), and how a dynamically scaled MLE anchor on exemplars prevents collapse when the pseudo-teacher is imperfect.
3. With only 4 chain-of-thought exemplars and answer only QA pairs, AWDPO achieves over 90% of the gains of a fully supervised CoT fine-tuning on GSM8K, and transfers reasoning capabilities to other math benchmarks, indicating an efficient and reproducible route to instill zero-shot reasoning in compact models.

## 2 RELATED WORK

**Aligning Language Models with Preferences.** Our work builds on Direct Preference Optimization (DPO) (Rafailov et al., 2023), a stable alternative to complex Reinforcement Learning from Human Feedback (RLHF) pipelines (Ouyang et al., 2022). While standard DPO uses binary preferences, we introduce a continuous-valued *advantage* weight to better capture the magnitude of improvement between outputs. Unlike typical DPO, which requires manually labeled preference pairs, our feedback signal comes from the model itself: we compare few-shot and zero-shot responses using ground-truth task answers. This eliminates the need for external preference annotation or a learned reward model, though it does assume access to correct answers.

**Distilling Chain-of-Thought Reasoning.** Our approach relates to knowledge distillation and in particular the distillation of reasoning processes from larger or stronger models into smaller ones. Chain-of-thought (CoT) prompting (Wei et al., 2022; Kojima et al., 2022) elicits step-by-step

reasoning from LLMs, dramatically improving accuracy on math and reasoning tasks. Several works have aimed to transfer these reasoning abilities to smaller models. Magister et al. (2022) and Li et al. (2023) use a powerful LLM to generate rationales for each training query and fine-tune a smaller model on these rationales, finding that even 1.3B parameter models can benefit from CoT supervision. Wang et al. (2023) propose SCOTT, which distills a large model's *self-consistent* reasoning by using multiple sampled chains of thought and training the student to reproduce the most consistent answers. Other strategies include filtering or chunking rationales to avoid overwhelming the student (Zelikman et al., 2022; Qin et al., 2023). Unlike teacher-student approaches that assume the student can follow instructions, we perform self-distillation on base models that lack instruction-following capabilities entirely. A related idea is STaR (Zelikman et al., 2022), where a model iteratively learns from its own generated solutions through repeated cycles of generation, filtering, and supervised fine-tuning. While STaR's goal is iterative self-improvement, AWDPO focuses on a different problem: directly distilling a model's latent few-shot reasoning capabilities into its zero-shot distribution. Rather than STaR's iterative approach that filters for correct solutions, our method uses single-pass preference optimization to learn from both correct and incorrect model outputs simultaneously. By weighting these pairs by their advantage, our approach directly internalizes the reasoning scaffolding provided by few-shot prompts, making it more lightweight and efficient than multi-round pipelines.

**Parameter-Efficient Fine-Tuning and Regularization.** We adopt *parameter-efficient fine-tuning* in the form of LoRA (Hu et al., 2022) which has been shown to achieve on-par performance with full fine-tuning on various NLP tasks while modifying only a tiny fraction of parameters. Here, we find LoRA not only provides efficiency but also helps enforce a trust region: by capping the update directions, the fine-tuned model remains closer to the original.

## 3 ADVANTAGE-WEIGHTED PREFERENCE LOSS

Direct Preference Optimization (DPO) (Rafailov et al., 2023) provides a framework for tuning a policy $\pi_\theta$ using preferences between pairs of outputs. We treat the model's own few-shot output $y^+$ as preferred over its zero-shot output $y^-$. AWDPO modifies the DPO objective by introducing a continuous advantage weight, $A(x) = R(x, y^+) - R(x, y^-)$, where $R(x, y)$ is a scalar reward function. This converts binary preferences into graded optimization signals based on the quality gap between the two outputs. Our final loss function is:

$$\mathcal{L}_{\text{AWDPO}}(\theta) = -\mathbb{E}_{x, y^+, y^-}[A(x) \log \sigma(\beta \Delta_\theta(x))]$$

where $\Delta_\theta(x)$ is the difference in log-probabilities between the policy and a reference model for the pair $(y^+, y^-)$.

### 3.1 THEORETICAL PERSPECTIVE

We justify AWDPO by showing it can be derived as a KL-constrained optimization problem and is stabilized by implicit trust regions.

**AWDPO as Projected DPO.** AWDPO arises from a constrained preference optimization objective.

**Theorem 1.** *The optimal policy for maximizing the advantage-weighted objective subject to a KL constraint against a reference policy $\pi_{ref}$ takes the form $\pi^*(y|x) \propto \pi_{ref}(y|x) \exp(\frac{1}{\beta}(A(x) \cdot (\mathbf{1}[y = y^+] - \mathbf{1}[y = y^-])))$ (Proof: see Appendix B).*

This form shows that the optimal policy is the reference policy reweighted by an exponential factor that boosts $y^+$ and suppresses $y^-$ in proportion to the advantage $A(x)$. In other words, AWDPO exponentially tilts the reference distribution toward the preferred output. Substituting this form into the preference model yields the AWDPO loss. While derived with a reference model, our implementation does not require one; stability is enforced via the implicit trust regions described next.

**Implicit Trust Regions.** We use two mechanisms to regularize training. First, we analyze LoRA updates as an implicit trust region.

**Theorem 2.** *Under smoothness assumptions, LoRA updates with rank $r$ are bounded such that $D_{KL}(\pi_{W+\Delta W}|\pi_W) \leq L\|\Delta W\|_F \leq L\sqrt{r}c_A c_B$ (Proof: see Appendix B).*

Intuitively, lower-rank updates constrain $\|\Delta W\|_F$, so small $r$ enforces a tighter trust region around the base model – a key reason LoRA stabilizes AWDPO training.

We use an MLE anchor on correct few-shot examples as a "teacher trust region."

**Theorem 3.** *The MLE loss term, $\mathcal{L}_{MLE}$, minimizes the KL divergence between the model's few-shot distribution and the empirical distribution of correct examples (Proof: see Appendix B).*

**Dynamic Loss Balancing.** To prevent either the preference loss or the MLE anchor from dominating, we dynamically scale the MLE term: $\lambda_{\text{dyn}} = \frac{\mathcal{L}_{\text{AWDPO}}}{\mathcal{L}_{\text{MLE}}+\varepsilon}$.

**Proposition 1.** *This dynamic weighting ensures that the gradient magnitudes of both loss components remain balanced during training.*

A full explanation of dynamic loss balancing can be found in Appendix B

### 3.2 REWARD FUNCTION

Our method uses a compound, rules-based reward function rather than a learned one. It primarily rewards the correctness of the final answer, with smaller bonuses for proper output formatting and heuristics that encourage stepwise reasoning. A detailed breakdown of the reward components is provided in Appendix C.

## 4 TRAINING ALGORITHM

The complete AWDPO training process is summarized in Algorithm 1. It outlines the steps for sampling generation pairs, computing the advantage-weighted losses, and updating the model.

---

**Algorithm 1** Advantage-Weighted Direct Preference Optimization (AWDPO)

---

1: **Input:** Base policy $\pi_\theta$, questions $\{x_i\}$, $K$ exemplars, reward function $R$.
2: **for** each question $x_i$ **do**
3:     Sample few-shot output $y_i^+ \sim \pi_\theta(\cdot|\text{prompt}(x_i, K))$.
4:     Sample zero-shot output $y_i^- \sim \pi_\theta(\cdot|x_i)$.
5:     Compute advantage $A_i = R(x_i, y_i^+) - R(x_i, y_i^-)$.
6:     Compute DPO logit $\Delta_i = \log \frac{\pi_\theta(y_i^+|x_i)}{\pi_\theta(y_i^-|x_i)}$.
7:     Compute preference loss $\mathcal{L}_{\text{AWDPO},i} = -A_i \log \sigma(\beta\Delta_i)$.
8:     **if** $y_i^+$ is a correct exemplar trace **then**
9:         Compute anchor loss $\mathcal{L}_{\text{MLE},i} = -\log \pi_\theta(y_i^+|\text{prompt}(x_i, K))$.
10:     **end if**
11: **end for**
12: Compute total losses $\mathcal{L}_{\text{AWDPO}}, \mathcal{L}_{\text{MLE}}$.
13: Set dynamic weight $\lambda_{\text{dyn}} = \frac{\mathcal{L}_{\text{AWDPO}}}{\mathcal{L}_{\text{MLE}}+\varepsilon}$.
14: Compute final objective $\mathcal{L}_{\text{total}} = \mathcal{L}_{\text{AWDPO}} + \lambda_{\text{dyn}}\mathcal{L}_{\text{MLE}}$.
15: Update parameters $\theta$ by descending the gradient $\nabla_\theta \mathcal{L}_{\text{total}}$.

---

Several aspects of Algorithm 1 warrant brief elaboration. Our objective handles both positive and negative advantages (line 6), allowing the model to learn even when the zero-shot output occasionally outperforms the few-shot one. A key detail of our method is how log-probabilities are calculated for each loss component. For the AWDPO loss, the log-probabilities of both the few-shot response ($y^+$) and the zero-shot response ($y^-$) are calculated conditioned on the zero-shot prompt. This trains the model's zero-shot distribution to favor the kind of structured text elicited by few-shot prompting. In contrast, the MLE anchor loss is calculated for correct few-shot responses conditioned on their original few-shot prompts, which ensures that the model's ability to reason when prompted (i.e., the "teacher") does not degrade. Finally, note that while our theoretical derivation uses a reference model,

our final algorithm forgoes one, as stability is maintained by the implicit trust regions from LoRA and the MLE anchor.

## 5 EXPERIMENTS

We evaluate AWDPO on mathematical reasoning tasks, focusing primarily on **GSM8K** (Cobbe et al., 2021), a benchmark of grade-school math word problems. We also test cross-task generalization to basic math (**SVAMP** (Patel et al., 2021) and **ASDiv** (Miao et al., 2020)), advanced math (**MATH-500** (Hendrycks et al., 2021; Lightman et al., 2024), as well as three datasets drawn from the Big Bench Hard benchmark (Suzgun et al., 2022). Our base models are **Qwen-2.5** language models of 0.5B, 1.5B, and 3B parameters,[1] which initially lack the ability to act as coherent chatbots like their instruction-tuned counterparts.

**Baselines.** We compare AWDPO against the following: (1) **Few-shot prompting**: providing 4 exemplars (the same ones used by AWDPO) as demonstrations at inference time, but no model fine-tuning. This represents the base model's upper-bound performance using prompting alone. (2) **Vanilla DPO**: fine-tuning using the DPO objective of Rafailov et al. (2023) on the same data. We use 6k training pairs derived from Qwen 2.5 3B-Instruct. (3) **Unweighted preference (Filtered-DPO)**: a variant of the above where we still generate pairs $(y^+, y^-)$ but calculate both our preference and MLE loss only over the correct reasoning traces. In practice, this is very similar to online DPO with an MLE term, but it is included here for the sake of demonstrating the difference between training only on unweighted correct responses and on all responses weighed by reward. (4) **Supervised Fine-Tuning (SFT)**: an upper bound where we fine-tune the model on the full GSM8K training set with chain-of-thought solutions generated by Qwen 2.5 3B-Instruct (7,473 chain-of-thought examples). This originates from the same dataset from which we draw few-shot exemplars and preference pairs for vanilla DPO. SFT represents the best achievable performance with extensive labeled data and serves as an oracle for comparison. We do not include the zero-shot performance of the base models due to the fact that they lack any kind of instruction tuning and therefore struggle to generate coherent, easily gradeable solutions.

Our fine-tuning strategy is tailored to the requirements of each method. Our proposed method, **AWDPO**, and its direct ablation, **Filtered-DPO**, both use Low-Rank Adaptation (LoRA; (Hu et al., 2022)) with a rank of $r = 64$, which acts as a lightweight regularizer and implicitly stabilizes optimization. In contrast, the **SFT** baseline is fully fine-tuned as it requires no such regularization, and the standard **DPO** baseline is also fully fine-tuned, as it maintains stability via its built-in KL divergence from a reference model. For a fair comparison of training efficiency, all methods are trained for the same number of epochs. To increase prompt diversity for our methods, we randomly sample $k \in \{2, 3, 4\}$ few-shot exemplars for each training instance. Further hyperparameter details can be found in Appendix D.

### 5.1 RESULTS ON GSM8K

Table 1 presents the accuracy (pass@1) on the GSM8K test set for each method and model size. We report the percentage of problems answered correctly (with the correct final numeric answer) by each model. For methods producing chain-of-thought (few-shot prompting, AWDPO, SFT, etc.), a solution is considered correct if the final answer extracted from the chain-of-thought is correct.

As seen in Table 1, AWDPO dramatically boosts the reasoning performance of small models on these basic math benchmarks. For the 0.5B model, AWDPO reaches 38.8% accuracy, recovering about 94.4% of the gains of the fully-supervised model (41.1%). For the 1.5B model, AWDPO achieves 64.5%, which is 92.2% of the supervised model's 69.9%. In the case of the 3B model, it recovers 95% of the fine-tuned performance (81.75%). In absolute terms, over its few shot prompted performance, the AWDPO trained 0.5B model achieves a $+14.3$ point improvement, the 1.5B model achieves $+8.45$ improvement, and the 3B model achieves a $+23.85$ point improvement.

The standard DPO baseline did not yield coherent reasoning improvements on base models. This highlights a key challenge: existing preference-based methods are designed primarily for aligning

---

[1]These are smaller variants of Qwen (Bai et al., 2023), a multilingual transformer LM. We use the English-oriented versions for fair comparison.

| Method | Model | pass@1 (%) | Chain-of-Thought Examples |
|--------|-------|-----------|---------------------------|
| *0.5B Model* | | | |
| DPO | Qwen-2.5 0.5B | 0.00 | 6,921 |
| Filtered-DPO | Qwen-2.5 0.5B | 13.73 | **4** |
| 4-shot | Qwen-2.5 0.5B | 24.53 | **4** |
| AWDPO (Ours) | Qwen-2.5 0.5B | 38.83 | **4** |
| SFT | Qwen-2.5 0.5B | **41.13** | 7,473 |
| Answer-SFT | Qwen-2.5 0.5B | 5.35 | **4** |
| *1.5B Model* | | | |
| DPO | Qwen-2.5 1.5B | 0.00 | 6,921 |
| Filtered-DPO | Qwen-2.5 1.5B | 4.84 | **4** |
| 4-shot | Qwen-2.5 1.5B | 56.00 | **4** |
| AWDPO (Ours) | Qwen-2.5 1.5B | 64.45 | **4** |
| SFT | Qwen-2.5 1.5B | **69.89** | 7,473 |
| Answer-SFT | Qwen-2.5 1.5B | 10.24 | **4** |
| *3B Model* | | | |
| DPO | Qwen-2.5 3B | 0.00 | 6,921 |
| Filtered-DPO | Qwen-2.5 3B | 56.30 | **4** |
| 4-shot | Qwen-2.5 3B | 53.78 | **4** |
| AWDPO (Ours) | Qwen-2.5 3B | 77.63 | **4** |
| SFT | Qwen-2.5 3B | **81.75** | 7,473 |
| Answer-SFT | Qwen-2.5 3B | 14.40 | **4** |

Table 1: Performance comparison of training methods on GSM8K dataset, ordered by method. AWDPO refers to our proposed Advantage-Weighted DPO method. Best results per model size are shown in bold.

instruction-tuned models, whereas AWDPO addresses the distinct problem of distilling reasoning scaffolding from few-shot prompts into the zero-shot distribution of base models.

For "Filtered-DPO", the non-advantage weighted ablation, results are also lacking. On the 0.5B model, Filtered-DPO reaches 13.73%, which is far below AWDPO. On the 1.5B model, Filtered-DPO completely fails (only 4.84%). We found that the unweighted preference loss caused instability for the larger model, often driving it to output trivial or malformed answers for most questions (hence near-zero accuracy). This highlights the importance of the advantage weighting and the MLE anchor in AWDPO: without them, the training can over-optimize on noisy preferences and collapse.

Few-shot prompting remains a strong baseline, showing the latent reasoning ability of these models when scaffolding is provided. AWDPO's contribution is to internalize that scaffolding into the model weights, so it can perform comparably in a zero-shot setting without needing exemplars at inference. Moreover, few-shot prompting requires providing the exemplars at inference time for every query, which is inconvenient or infeasible in some scenarios (e.g. limited context window or API usage constraints). AWDPO yields a model that can perform well *without* any prompt exemplars, after a one-time fine-tuning.

As an additional test, we formulate a baseline called 'Answer-SFT' which comprises of fine-tuning each model on 7,473 answer-only pairs and the 4 chain-of-thought reasoning traces used as few-shot examples in AWDPO and Filtered-DPO. This evaluation is a direct comparison of SFT to AWDPO with identical data. This method effectively collapses to fine-tuning an LLM to produce only the answer without any intermediate reasoning steps. The approach does not perform well at any model size. This reinforces the advantage of the focal approach with similar dataset budgets.

Figure 1 illustrates these results. AWDPO closes most of the gap between few-shot and fully supervised performance, while baselines without advantage or without fine-tuning fall short.

## 5.2 Zero-Shot Generalization to Other Datasets

To test whether AWDPO transfers few-shot reasoning patterns into zero-shot performance beyond GSM8K, we evaluate GSM8K-trained AWDPO models on SVAMP and ASDiv without additional training or exemplars. We compare against fully supervised GSM8K-tuned models (SFT) of the same size to assess out-of-distribution generalization. Table 2 shows these results.

For the 1.5B model, AWDPO reaches 69.1% on SVAMP and 86.2% on ASDiv, which is 85–91% of the supervised model's accuracy on those sets. In other words, in cross-dataset generalization, the

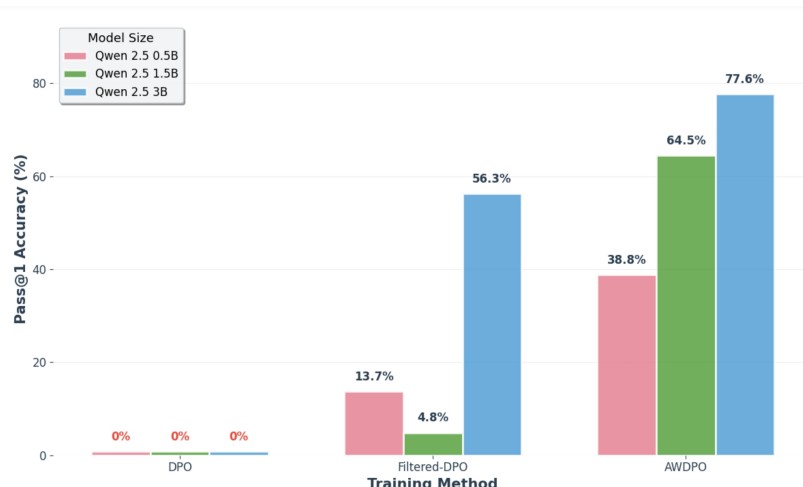

Figure 1: **Mathematical reasoning accuracy on GSM8K** for three model sizes (0.5B, 1.5B, and 3B) across different training methods. AWDPO (ours) achieves performance very close to the upper-bound supervised fine-tuning (SFT) while using minimal labeled data, and significantly outperforms both the base model's zero-shot/few-shot abilities and a standard DPO fine-tuning.

| Model | Method | SVAMP | ASDiv | GSM8K | MATH-500 |
|-------|--------|-------|-------|-------|----------|
|       |        | p@1   | p@1   | p@1   | p@1      |
| 0.5B  | SFT    | 0.543 | 0.826 | 0.411 | 0.151    |
|       | AWDPO  | 0.408 | 0.703 | 0.388 | 0.112    |
| 1.5B  | SFT    | 0.805 | 0.941 | 0.699 | 0.295    |
|       | AWDPO  | 0.691 | 0.862 | 0.645 | 0.268    |
| 3B    | SFT    | 0.888 | 0.961 | 0.818 | 0.373    |
|       | AWDPO  | 0.791 | 0.866 | 0.776 | 0.324    |

Table 2: Pass@1 performance of AWDPO vs supervised fine-tuning (SFT) on math reasoning benchmarks (SVAMP, ASDiv, GSM8K, and MATH-500).

AWDPO model is close to on par with the large-data fine-tuned model. For the 0.5B model, the gap is larger (e.g., 40.9% vs 54.3% on SVAMP), but AWDPO still attains about 75% of the supervised model's transfer performance. The 3B model recovers the most performance, achieving 89% of the fine-tuned performance on SVAMP and 90% on ASDiv.

On the harder competition-level math in MATH-500, overall performance is lower for both SFT and AWDPO, but AWDPO still recovers a substantial portion of supervised accuracy. At 1.5B and 3B, it reaches about 90.8% and 86.8%, respectively, of the supervised model's performance. This indicates that AWDPO transfers beyond word-problem style reasoning to more abstract, multi-step competition problems.

To further probe the generalization capabilities of our method on non-mathematical reasoning, we also evaluated our models on three challenging subtasks from the Big Bench Hard (BBH) benchmark (Suzgun et al., 2022). We found that while performance was more varied than on mathematical datasets, AWDPO showed encouraging results, particularly at the 3B scale where it approached or exceeded the SFT baseline on several tasks. This suggests the reasoning patterns it internalizes have broader applicability, though further investigation is warranted. The full results and a detailed analysis are provided in Tables 5, 6, and 7 in Appendix A.

Taken together, these results show that AWDPO effectively distills few-shot reasoning into zero-shot capabilities, yielding strong generalization across datasets and difficulty levels, while also exhibiting encouraging—if uneven—transfer to non-math reasoning tasks.

The supervised model saw 7k chain-of-thought GSM8K solutions (many overlapping SVAMP/ASDiv patterns), whereas AWDPO used only four CoT exemplars plus answer-only examples. The fact that AWDPO closes most of the gap is evidence of effective generalization from those few examples plus self-generated training signals.

# 6 ANALYSIS AND ABLATIONS

We perform additional analyses to understand AWDPO's behavior and the importance of its components.

**Teacher Quality Impact** We investigated AWDPO's robustness to teacher quality by evaluating on Llama 3.2 3B (AI at Meta, 2024), which exhibits substantially weaker baseline reasoning than the Qwen models.

- We observe that AWDPO successfully bootstraps reasoning even from severely degraded teachers. Starting from a few-shot baseline of less than 1%, AWDPO was able to improve Llama 3.2 3B to 28.81% on GSM8K, a more than 400x performance improvement.

- Llama 3.2 3B shows a performance pattern similar to the Qwen models, with AWDPO outperforming Filtered-DPO, standard DPO, 4-shot prompting the base model, and fine-tuning on 7,473 answer-only observations with only 4 fixed chain-of-thought examples.

- We also observe that when compared to SFT, Llama 3.2 3B has a more substantial performance gap than any of the Qwen series of models. This implies that although it is capable of bootstrapping reasoning out of small models, the final performance is bounded by the base model's capabilities.

- Despite lower absolute performance, the pattern holds: AWDPO > Answer-SFT > Filtered-DPO > few-shot prompting, validating the method's core mechanism across model families. The AWDPO approach shows consistent relative gains across architectures. In sum, AWDPO successfully bootstraps reasoning even from severely degraded teachers.

These results demonstrate that AWDPO's advantage-weighted learning works across architectures with consistent relative improvements. A full breakdown of performance across methods for Llama 3.2 3B can be found in Table 3.

| Method | Model | pass@1 (%) | Chain-of-Thought Examples |
|---|---|---|---|
| DPO | Llama 3.2 3B | 0.00 | 6,921 |
| Filtered-DPO | Llama 3.2 3B | 2.11 | **4** |
| 4-shot | Llama 3.2 3B | 0.07 | **4** |
| AWDPO (Ours) | Llama 3.2 3B | 28.81 | **4** |
| SFT | Llama 3.2 3B | **54.63** | 7,473 |
| Answer-SFT | Llama 3.2 3B | 8.78 | **4** |

Table 3: Performance of training methods on GSM8K dataset for Llama 3.2 3B, ordered by method. AWDPO refers to our proposed Advantage-Weighted DPO method. Best results per model size are shown in bold.

**Impact of Advantage Weighting.** We ablated the advantage in our loss by training a variant with $A(x)$ clamped to 1 for all $x$ (equivalent to an online DPO setup with an MLE anchor for the few-shot distribution). As shown earlier, this resulted in significantly lower performance and even divergence for the larger model. Specifically, removing advantage weighting prevented the model from reliably transferring the reasoning scaffolding of few-shot prompts into its zero-shot distribution, often leading to collapse. On the 0.5B model, performance initially rose but then oscillated heavily; on the 1.5B model, training eventually collapsed with the reward dropping sharply; and even at 3B, where results were somewhat steadier, the trajectory flattened and degraded toward the end. In contrast, AWDPO maintained steady improvements at every scale. These patterns, consistent across the three ablation

| Model | Rank | $\alpha$ | SVAMP@1 | ASDiv@1 | GSM8K@1 | M-500@1 | Avg@1 | KL | $\text{KL}_{\text{std}}$ |
|---|---|---|---|---|---|---|---|---|---|
|  | SFT | SFT | 0.878 | 0.961 | 0.807 | 0.368 | 0.753 | — | — |
|  | 64 | 64 | 0.767 | 0.876 | 0.749 | 0.349 | 0.685 | 0.007 | 0.040 |
| 3B | 64 | 128 | 0.833 | 0.931 | 0.760 | 0.357 | 0.720 | 0.021 | 0.171 |
|  | 128 | 128 | 0.690 | 0.790 | 0.731 | 0.288 | 0.625 | 0.017 | 0.085 |
|  | 64 | 256 | 0.818 | 0.877 | 0.736 | 0.345 | 0.694 | 0.232 | 0.999 |
|  | SFT | SFT | 0.790 | 0.941 | 0.691 | 0.318 | 0.684 | — | — |
|  | 64 | 64 | 0.691 | 0.862 | 0.631 | 0.258 | 0.610 | 0.013 | 0.069 |
| 1.5B | 32 | 64 | 0.676 | 0.875 | 0.626 | 0.251 | 0.607 | 0.026 | 0.186 |
|  | 64 | 128 | 0.701 | 0.839 | 0.629 | 0.263 | 0.608 | 0.086 | 0.500 |
|  | 128 | 128 | 0.778 | 0.897 | 0.641 | 0.267 | 0.646 | 0.066 | 0.421 |
|  | SFT | SFT | 0.539 | 0.817 | 0.40 | 0.143 | 0.474 | — | — |
|  | 64 | 64 | 0.408 | 0.703 | 0.334 | 0.104 | 0.387 | 0.281 | 0.831 |
| 0.5B | 32 | 64 | 0.427 | 0.735 | 0.346 | 0.134 | 0.411 | 0.250 | 0.811 |
|  | 16 | 32 | 0.384 | 0.581 | 0.366 | 0.107 | 0.359 | 0.094 | 0.320 |
|  | 32 | 32 | 0.450 | 0.672 | 0.375 | 0.114 | 0.403 | 0.074 | 0.263 |

Table 4: Zero-shot pass@1 and KL diagnostics across model scales and LoRA settings. For the sake of simplicity, all ablations were conducted on a checkpoint taken after 1,000 training iterations.

plots, highlight that advantage weighting is not only beneficial but essential for stabilizing training. Visualizations of these dynamics can be seen in Appendix D.

**Effect of Dynamic Loss Balancing.**    We tried using a fixed weight for the MLE anchor term (tuned via grid search) and found that dynamic balancing was critical for stabilizing the self-distillation process, since the few-shot teacher signal is sparse and noisy. If the weight is too low, the model sometimes collapses to copying the few-shot outputs even when they are wrong (since nothing pulls it back). If too high, the model sticks too closely to exemplars and fails to fully leverage the preference signal (yielding lower improvement). Our auto-scaling strategy ensured that $\mathcal{L}_{\text{AWDPO}}$ and $\mathcal{L}_{\text{MLE}}$ remained of comparable scale. An interesting side effect was that the ratio of these losses self-adjusted differently for the 0.5B vs 1.5B model: the larger model put slightly more relative emphasis on the anchor (perhaps because it could memorize the 4 exemplars more easily, it needed a stronger anchor to not overfit preferences). Overall, this component greatly simplified training: we did not need to retune any hyperparameter when scaling up model size or changing the number of exemplars.

**Significance of Capacity-Aware KL Tuning.**    These ablations reveal four key insights for applying AWDPO effectively:

1. **Capacity-aware trust regions.** Although OOD pass@1 accuracy follows a "Goldilocks" curve in token-level KL divergence, the *optimal* KL shifts upward as model size decreases: $\approx 0.02$ nats for 3 B, $\approx 0.06$ nats for 1.5 B, and $\approx 0.25$ nats for 0.5 B (Table 4). Smaller models require proportionally larger LoRA updates (higher $\alpha/r$) to internalize few-shot reasoning, whereas larger models only need a gentle adjustment.

2. **KL as a universal tuning dial.** Token-level KL between the fine-tuned and base policies serves as a *predictive proxy* for OOD performance: models with KL below or above their size-specific sweet spot underperform on SVAMP/ASDiv. By measuring KL on a small probe set, practitioners can anticipate whether their AWDPO run will over-specialize (KL too large) or under-express the reasoning signal (KL too small).

3. **Lean hyperparameter recipe.** Instead of exhaustive grid searches, one can target the empirically validated KL band by: (a) selecting $\alpha/r$ to approximate the desired KL for that model size, (b) running a brief adaptation pass, and (c) adjusting $\alpha$ (or LoRA rank) up or down until the measured KL lands in the optimal range. This procedure consistently yields 70–95% of fully supervised transfer performance using only four exemplars.

4. **Robust zero-shot generalization.** When tuned to its capacity-specific trust region, AWDPO achieves a large fraction of the supervised fine-tune's accuracy on both in-domain (GSM8K)

and out-of-domain (SVAMP, ASDiv) benchmarks, despite using 1/1750th of the chain-of-thought data. This demonstrates that simple self-distillation via advantage-weighted preference losses plus lightweight LoRA regularization can reliably compress reasoning skills into compact LMs without heavy annotation or RL.

# 7 CONCLUSION

We presented Advantage-Weighted Direct Preference Optimization (AWDPO), a novel method for aligning a language model with its own chain-of-thought reasoning ability. By using the model's few-shot outputs as a pseudo-teacher and weighting preference learning by advantage, AWDPO achieves a form of self-distillation that drastically reduces the need for supervised data. Our experiments on math reasoning tasks demonstrated that with only 4 exemplars and answer only QA pairs, small models can attain over 90% of the performance of a fully supervised fine-tune, and even generalize those gains to new datasets in zero-shot settings. Key to our approach is the use of implicit regularization (LoRA adapters and an MLE anchor) and dynamic loss balancing, which together ensure stable and effective training without trial-and-error hyperparameter tuning.

AWDPO's success opens several avenues for future work. One direction is to apply this method to other domains of reasoning or decision-making, such as code generation or logical proofs, where few-shot prompting helps but large-scale chain-of-thought data is scarce. Another direction is to explore richer forms of reward $R(x, y)$ for advantage computation—potentially incorporating heuristic checks or external validators to refine the signal without human labels. Finally, AWDPO suggests that even in the absence of a human feedback loop, language models can *bootstrap* their capabilities; integrating this idea with human-in-the-loop training (e.g. to correct the pseudo-teacher when it is wrong) could further amplify the benefits. We hope this work spurs research on lightweight alignment that relies on a model's own knowledge, bringing advanced reasoning to smaller models.

# 8 LIMITATIONS

We do not include a full multi-round STaR baseline due to compute and time constraints. Instead, we compare against a Filtered-DPO variant that captures the same hard-filtering mechanism, highlighting the distinction between binary filtering approaches and our continuous, advantage-weighted formulation.

Our experiments are also limited in scope. We primarily train and evaluate on GSM8K, with additional cross-dataset evaluation but no full training runs on other benchmarks. In addition, while AWDPO consistently improves zero-shot reasoning relative to vanilla DPO and self-training baselines, it does not surpass strong supervised fine-tuning on curated chain-of-thought demonstrations. This highlights that our method is best viewed as a data-efficient alternative when high-quality chain-of-thought data is scarce, rather than a universal replacement for SFT.

Additionally, AWDPO assumes a baseline teacher that can generate meaningful training examples from few-shot examples. While it does show the ability to bootstrap reasoning from an unreliable few-shot teacher distribution, as in the case of Llama 3.2 3B, it is limited in comparison to SFT when the base model itself is relatively weak. Finally, AWDPO assumes access to simple correctness-based reward signals; extending the approach to weaker or learned reward models is left to future work.

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

## ETHICS STATEMENT

This paper aims to advance research in machine learning. Our study uses publicly available benchmark datasets and does not involve the collection or processing of personally identifiable information. We do not target sensitive attributes or deploy the method in user-facing settings. We see no specific ethical concerns that require additional disclosure beyond these general considerations.

## LLM USE DISCLOSURE

During the preparation of this paper, we used an LLM (GPT-5) as a writing assistant for grammar, sentence structure, and phrasing. The model did not generate technical content, code, analyses, or results. No proprietary, personal, or sensitive data were shared with the tool. All ideas, methods, and experimental results were created and verified by the authors, who take full responsibility for the paper.

## REPRODUCIBILITY STATEMENT

We will provide the source code and configuration for the key experiments, including instructions on how to generate data and train the models. All proofs are stated in the appendix with explanations and underlying assumptions. We thoroughly checked the implementation and also verified that the results from the proposed models hold. All reported numbers in the paper can be reproduced using the released code, configuration files, and instructions.

APPENDIX

# A   PERFORMANCE ON BBH

## A.1   ZERO-SHOT PERFORMANCE TRANSFER

AWDPO exhibits greater variability on Big Bench Hard tasks than on mathematical reasoning benchmarks, reflecting the brittleness of symbolic and logical reasoning where single-token errors often lead to failure.

**Experimental setup:** All models in this section were trained exclusively on GSM8K using AWDPO, then evaluated *zero-shot* on three BBH reasoning tasks (Boolean Expressions, Causal Judgment, Logical Deduction). No BBH-specific training or exemplars were used. This tests whether reasoning patterns learned from mathematical problem-solving transfer to other domains.

Tables 5, 6, and 7 present comprehensive results across all LoRA configurations tested for each model size.

**Key Observations Across Model Scales**

- Scale-dependent transfer: The gap between AWDPO and SFT narrows substantially with model size. At 0.5B, AWDPO underperforms significantly (e.g., 0.24 vs 0.53 on Boolean p@1). At 1.5B, certain configurations approach SFT performance (e.g., 0.443 vs 0.527 on Causal p@1). At 3B, AWDPO achieves near-parity on Boolean reasoning (0.727 vs 0.759 at p@1) and notably *exceeds* SFT on Logical Deduction (0.083 vs 0.043 at p@1).

- Self-consistency benefits: A consistent pattern emerges across all scales: while AWDPO lags SFT at p@1, the gap narrows considerably at p@5. For instance, at 1.5B on Causal reasoning, AWDPO achieves 0.880 at p@5 (97% of SFT's 0.908), despite only reaching 0.443 at p@1 (84% of SFT). This suggests AWDPO internalizes relevant reasoning patterns but benefits from sampling multiple solutions to surface correct answers reliably.

- Configuration sensitivity and KL correlation: Non-math reasoning performance correlates with KL divergence following the capacity-aware patterns identified in Section 6. Configurations with KL values far from the optimal range for each model size (0.02 nats for 3B, 0.06 nats for 1.5B, 0.25 nats for 0.5B) show degraded performance. For example, at 1.5B, r=64 $\alpha$=128 (KL=0.086) substantially underperforms r=64 $\alpha$=64 (KL=0.013).

**Interpretation**   These comprehensive results demonstrate that reasoning patterns learned from mathematical problem-solving via AWDPO transfer to non-mathematical domains, though with task- and scale-dependent effectiveness. The method works best when: (1) model capacity is sufficient (1.5B+), and (2) hyperparameters are tuned to capacity-appropriate KL ranges. The strong p@5 performance across all scales suggests AWDPO captures underlying reasoning structures but benefits from self-consistency decoding in brittle domains.

Our primary contribution targets mathematical reasoning in small models, where AWDPO recovers 90%+ of supervised performance. These zero-shot BBH results provide preliminary evidence of cross-domain transfer while honestly delineating where the method excels (sequential reasoning tasks like mathematics and causal judgment) and where further investigation is needed (pure symbolic manipulation).

# B   THEORETICAL PROOFS AND DERIVATIONS

## B.1   PROOF OF THEOREM 1

**Theorem 1.** *The optimal policy for maximizing the advantage-weighted objective subject to a KL constraint against a reference policy $\pi_{ref}$ takes the form $\pi^*(y|x) \propto \pi_{ref}(y|x) \exp(\frac{1}{\beta}(A(x) \cdot (\mathbf{1}[y = y^+] - \mathbf{1}[y = y^-])))$.*

| Method | Config | Boolean | | Causal | | Logical | | KL (nats) |
|--------|--------|---------|---------|--------|---------|---------|---------|-----------|
| | | p@1 | p@5 | p@1 | p@5 | p@1 | p@5 | |
| SFT | SFT | 0.759 | 0.976 | 0.558 | 0.875 | 0.043 | 0.165 | — |
| AWDPO | r=64, $\alpha$=64 | 0.635 | 0.943 | 0.320 | 0.724 | 0.064 | 0.250 | 0.007 |
| | r=64, $\alpha$=128 | **0.727** | **0.958** | 0.428 | 0.821 | **0.083** | **0.303** | 0.021 |
| | r=128, $\alpha$=128 | 0.668 | 0.946 | **0.492** | **0.882** | 0.053 | 0.212 | 0.017 |
| | r=64, $\alpha$=256 | 0.695 | 0.947 | 0.404 | 0.784 | 0.036 | 0.139 | 0.232 |

Table 5: Pass@1 and Pass@5 performance of AWDPO vs supervised fine-tuning (SFT) on Big Bench Hard subsets for Qwen 2.5 3B. AWDPO shows variable performance across configurations. KL divergence from base model is provided for each AWDPO configuration.

| Method | Config | Boolean | | Causal | | Logical | | KL (nats) |
|--------|--------|---------|---------|--------|---------|---------|---------|-----------|
| | | p@1 | p@5 | p@1 | p@5 | p@1 | p@5 | |
| SFT | SFT | 0.703 | 0.959 | 0.527 | 0.908 | 0.210 | 0.584 | — |
| AWDPO | r=64, $\alpha$=64 | 0.198 | 0.633 | **0.443** | **0.880** | 0.203 | **0.611** | 0.013 |
| | r=32, $\alpha$=64 | 0.171 | 0.559 | 0.401 | 0.849 | 0.136 | 0.474 | 0.026 |
| | r=64, $\alpha$=128 | 0.163 | 0.550 | 0.160 | 0.528 | 0.082 | 0.323 | 0.086 |
| | r=128, $\alpha$=128 | **0.283** | **0.745** | 0.436 | 0.864 | **0.093** | 0.345 | 0.066 |

Table 6: Pass@1 and Pass@5 performance of AWDPO vs supervised fine-tuning (SFT) on Big Bench Hard subsets for Qwen 2.5 1.5B. AWDPO shows variable performance across configurations, with best results often approaching or exceeding SFT at Pass@5. KL divergence from base model is provided for each AWDPO configuration.

*Proof.* Following the standard DPO derivation (Rafailov et al., 2023), we form the Lagrangian:

$$L = \mathbb{E}_{x,y^+,y^-}\left[A(x)\log\frac{\pi(y^+|x)}{\pi(y^-|x)}\right] - \beta\mathbb{E}_x[D_{KL}(\pi(\cdot|x)\|\pi_{\text{ref}}(\cdot|x))]$$

Taking the functional derivative with respect to $\pi(y|x)$ and setting to zero:

$$\frac{\delta L}{\delta\pi(y|x)} = A(x)\cdot\frac{\mathbf{1}[y=y^+]-\mathbf{1}[y=y^-]}{\pi(y|x)} - \beta\log\frac{\pi(y|x)}{\pi_{\text{ref}}(y|x)} - \beta = 0$$

Solving for $\pi(y|x)$ yields the stated form. □

### B.2 PROOF OF THEOREM 2

We now formalize how LoRA updates provide implicit KL regularization.

**Assumption 1 (Local Smoothness).** In a neighborhood of the current weights $W$, the log-probability function $\log\pi_W(y|x)$ is $L$-Lipschitz continuous with respect to the Frobenius norm, for inputs $x$ from the training distribution.

**Assumption 2 (LoRA magnitude control).**

During training, LoRA factors satisfy $\|A\|_2 \leq c_A$ and $\|B\|_2 \leq c_B$ (enforced via per-step gradient clipping and weight-norm monitoring). This yields $\|\Delta W\|_F \leq \sqrt{r}\,c_A c_B$. This is consistent with initialization scaling and gradient clipping used in practice.

**Theorem 2.** *Under Assumptions 1–2, LoRA updates with rank $r$ satisfy:*

$$D_{KL}(\pi_{W+\Delta W}|\pi_W) \leq L\|\Delta W\|_F \leq L\sqrt{r}c_A c_B.$$

**Proposition 2 (Local trust region).** Under Assumptions 1–2, for steps taken during fine-tuning, $D_{\text{KL}}(\pi_{W+\Delta W}\,|\,\pi_W) \leq L\,\|\Delta W\|_F \leq L\sqrt{r}\,c_A c_B$.

| Method | Config | Boolean | | Causal | | Logical | | KL (nats) |
|--------|--------|---------|---------|---------|---------|---------|---------|-----------|
| | | p@1 | p@5 | p@1 | p@5 | p@1 | p@5 | |
| SFT | SFT | 0.53 | 0.932 | 0.495 | 0.838 | 0.093 | 0.376 | — |
| AWDPO | r=64, $\alpha$=64 | 0.243 | **0.712** | 0.066 | 0.218 | **0.015** | **0.065** | 0.28 |
| | r=32, $\alpha$=64 | 0.177 | 0.543 | **0.091** | **0.281** | 0.001 | 0.005 | 0.25 |
| | r=16, $\alpha$=32 | **0.248** | 0.680 | 0.028 | 0.131 | 0.0005 | 0.003 | 0.094 |
| | r=32, $\alpha$=32 | 0.078 | 0.312 | 0.021 | 0.101 | 0.003 | 0.015 | 0.074 |

Table 7: Pass@1 and Pass@5 performance of AWDPO vs supervised fine-tuning (SFT) on Big Bench Hard subsets for Qwen 2.5 0.5B. AWDPO shows variable performance across configurations. KL divergence from base model is provided for each AWDPO configuration.

*Proof.* For any $y, x$ from the training distribution, the local Lipschitz property gives:

$$|\log \pi_{W+\Delta W}(y|x) - \log \pi_W(y|x)| \leq L\|\Delta W\|_F$$

The KL divergence is bounded by:

$$\begin{aligned} D_{KL}(\pi_{W+\Delta W}|\pi_W) &= \mathbb{E}_{y\sim\pi_{W+\Delta W}}[\log \pi_{W+\Delta W}(y|x) \\ &\quad - \log \pi_W(y|x)] \\ &\leq L\|\Delta W\|_F \\ &\leq L\sqrt{r}c_A c_B \end{aligned}$$

where the last inequality follows from Assumption 2.

The key insight is that the low-rank factorization $\Delta W = AB$ provides implicit regularization through multiple mechanisms:

- The number of parameters is reduced from $mn$ to $(m+n)r$

- Standard initialization schemes naturally bound the operator norms: $\|A\|_2 = O(\sqrt{r/m})$ and $\|B\|_2 = O(1)$ for typical schemes

- The optimization trajectory remains bounded due to the reduced parameter space

In practice, this means $\|\Delta W\|_F$ remains small throughout training, providing the trust region behavior without explicit constraints. $\square$

**Remark.** The $\sqrt{r}$ factor in the bound explicitly shows how lower rank provides stronger implicit regularization. For small $r$, this provides a tight trust region around the pre-trained weights, which is key to LoRA's effectiveness.

### B.3 PROOF OF THEOREM 3

The MLE loss term serves as a "teacher trust region" for the policy's few-shot distribution that prevents drift from high-quality demonstrations.

**Theorem 3.** *The MLE loss term, $\mathcal{L}_{MLE}$, minimizes the KL divergence between the model's few-shot distribution and the empirical distribution of correct examples.*

*Proof.* Let $\mathcal{D}_{\text{correct}} = \{(x_i, y_i^+) : r(x_i, y_i^+) = 1\}$ be the set of correct few-shot examples. The MLE anchor ensures:

$$\mathbb{E}_{(x,y^+)\in\mathcal{D}_{\text{correct}}}[D_{KL}(\delta_{y^+}\|\pi_\theta(\cdot|x))]$$

decreases with $\lambda_{\text{dyn}}$.

The MLE loss $\mathcal{L}_{\text{MLE}} = -\mathbb{E}_{(x,y^+)\in\mathcal{D}_{\text{correct}}}[\log \pi_\theta(y^+|x)]$ is equivalent to minimizing the cross-entropy between the delta distribution $\delta_{y^+}$ and $\pi_\theta(\cdot|x)$, which directly minimizes the stated KL divergence. $\square$

### B.4 EXPLANATION OF DYNAMIC LOSS BALANCING

Our dynamic weighting scheme $\lambda_{\text{dyn}} = \frac{\mathcal{L}_{\text{AWDPO}}}{\mathcal{L}_{\text{MLE}} + \varepsilon}$ ensures gradient balance.

**Proposition 1.** *When $\mathcal{L}_{MLE} > 0$, the dynamic weighting ensures:*

$$\|\nabla_\theta \mathcal{L}_{\text{AWDPO}}\| \approx \|\lambda_{\text{dyn}} \nabla_\theta \mathcal{L}_{\text{MLE}}\|$$

*preventing either objective from dominating the updates.*

This balancing is crucial because it prevents the MLE term from overwhelming the preference signal when correct examples are rare, and prevents the preference term from ignoring teacher demonstrations when they are abundant.

## C  REWARD FUNCTION DETAILS

Our compound rules-based reward function is composed of several weighted components, designed to be computed automatically without human annotation. The components are:

1. **Correctness**: A binary reward (+1.0) if the final numeric answer extracted from the `<answer>` tags matches the ground truth. This is the dominant component of the reward.

2. **Format**: A small bonus (+0.1) for correctly using the XML-style `<thinking>`... `</thinking>` and `<answer>`... `</answer>` tags.

3. **Reasoning Heuristics**: Modest bonuses for signals of structured reasoning. This includes a small reward (+0.05) for the presence of stepwise markers (e.g., 'first', 'next', 'therefore') within the `<thinking>` block. This reward is capped to avoid encouraging unnecessary verbosity.

4. **Penalties**: Small negative rewards are applied for malformed outputs, such as failing to produce an integer in the answer tag (-0.1) or excessive repetition of phrases (-0.1).

## D  TRAINING DETAILS

Hyperparameter details for each GSM8K AWDPO training run are as follows:

- Batch Size: 6
- Training Epochs: 1
- Learning Rate: 1e-5
- Temperature: 0.2
- AdamW optimizer
- Cosine LR scheduler
- rank = 64
- $\alpha = 64$

All training was done on a single A100 80GB with an average of $\sim$7 hours per model

## E  TRAINING DYNAMICS

We verify that AWDPO is not just optimizing the proxy reward, but genuinely improving the model's zero-shot reasoning ability, consistent with its goal of internalizing few-shot scaffolding. Figure 3 plots the progression of the total reward $R$, and the accuracy reward on the training set, over the course of training for each model. We see that as training progresses, the reward obtained by the few-shot vs zero-shot gap decreases, and importantly the zero-shot accuracy on GSM8K also increases in tandem. The "accuracy reward" curve in the figure isolates the component of $R$ corresponding to final answer correctness; it closely tracks actual accuracy, indicating that our reward proxy is well-aligned

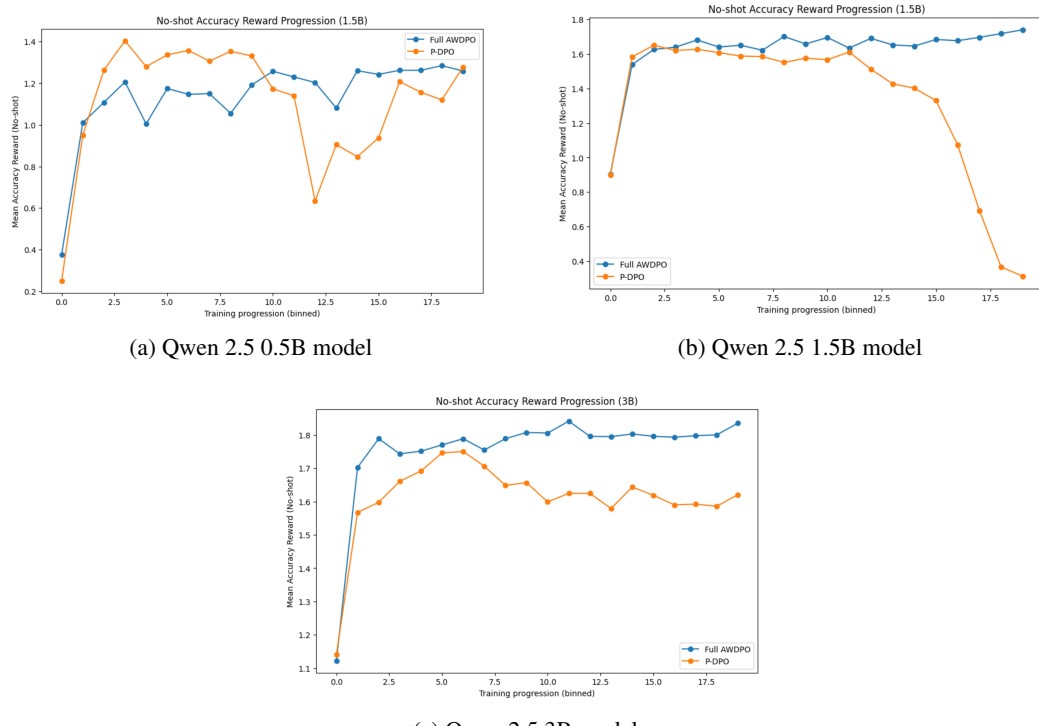

(a) Qwen 2.5 0.5B model          (b) Qwen 2.5 1.5B model

(c) Qwen 2.5 3B model

Figure 2: **Training Dynamics of the Filtered-DPO Baseline Across Model Scales.** Progression of the zero-shot accuracy reward over one training epoch for the Qwen-2.5 models. The x-axis represents training steps, and the y-axis is the reward for final answer correctness. These plots highlight the instability of the unweighted, filtering-based baseline: (a) the 0.5B model's performance oscillates heavily, (b) the 1.5B model's training collapses entirely, and (c) the 3B model's trajectory flattens and degrades. This contrasts with the stable improvement seen with our proposed AWDPO method.

with true task success. There is no sign of reward misoptimization (where reward would go up but accuracy stagnates or drops). This validates that AWDPO's training signal genuinely makes the model better at solving basic math problems.

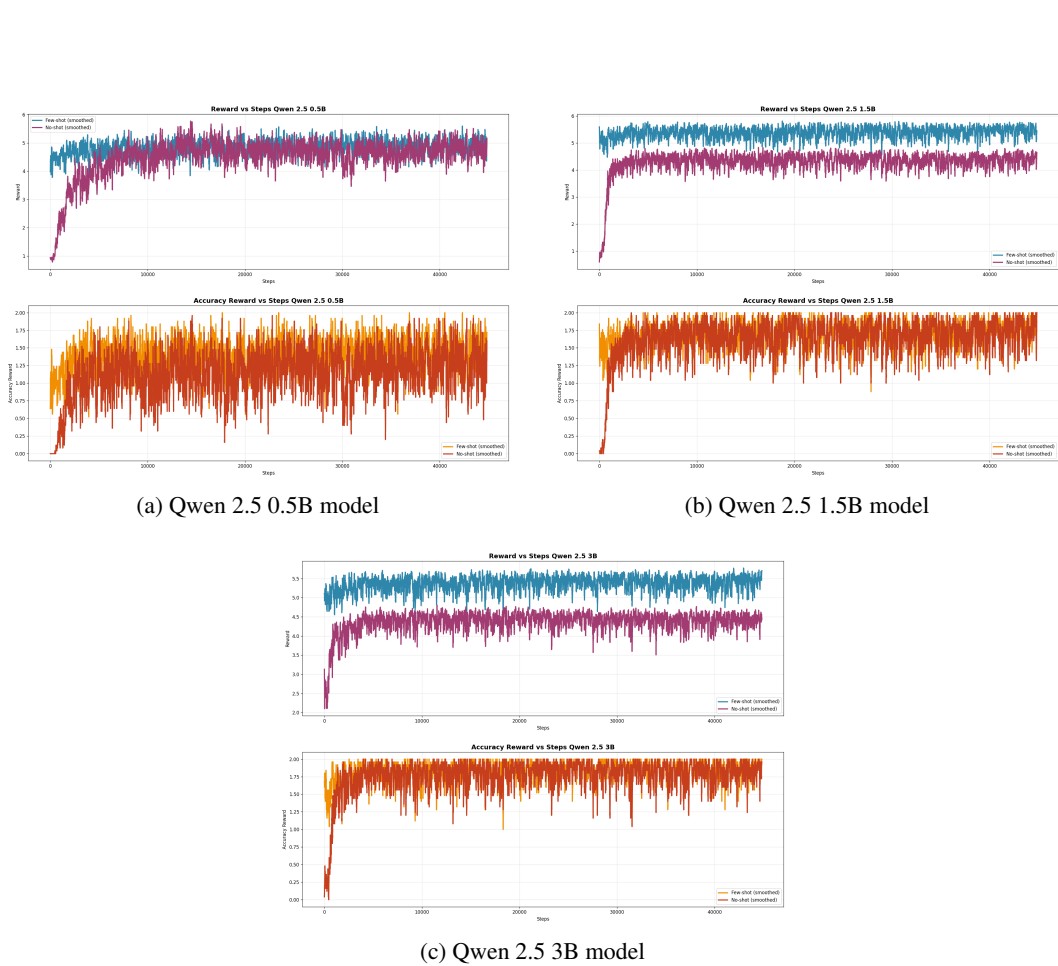

(a) Qwen 2.5 0.5B model

(b) Qwen 2.5 1.5B model

(c) Qwen 2.5 3B model

Figure 3: Reward progression of AWDPO Across Model Scales.

