# OpenReview forum: "Bootstrapping Zero-Shot Reasoning in Small Language Models via Advantage-Weighted Self-Distillation"
_ICLR.cc/2026/Conference — Submitted to ICLR 2026_

### Official Review · Reviewer_XGYQ · 2025-10-31

**Soundness:** 2
**Presentation:** 3
**Contribution:** 2
**Rating:** 6
**Confidence:** 3

**Summary:**

This paper introduces Advantage-Weighted Direct Preference Optimization (AWDPO), a data-efficient self-distillation method that bootstraps zero-shot reasoning in small language models by training them to prefer their own advantaged, few-shot-prompted outputs over their zero-shot ones.

**Strengths:**

- The AWDPO method proposed in this paper is intuitive and elegant. I particularly appreciate the approach of using the well-performing rule-based reward from RLVR as a weight for the DPO logits. The final form of the preference loss (Line 198) resembles the Policy Gradient in REINFORCE. I wonder if the authors have analyzed the potential connection between AWDPO and Policy Gradient?

- AWDPO performs well within the experimental scope covered in the paper. The DPO LoRA training, using self-distillation data collected from just 4-shot CoT golden responses, achieves results that approach the performance of full-parameter SFT trained on 7K+ golden responses.

**Weaknesses:**

- Although the authors acknowledge this limitation, I must point out that the experimental scope of the paper is very small. It only involves three small models from a single series (Qwen2.5) and are trained on only one data source (GSM8k). While I understand that expanding the scope would consume more computational resources, having both the number of model series and data sources limited to one weakens my confidence in the general usability of AWDPO.

- I would like to know in what scenarios we should use AWDPO to train our Base LLM.
    - If the objective is to save computational resources: AWDPO requires DPO LoRA training. For small models, how much memory and computational power does this save compared to full-parameter SFT? If the savings are not substantial, why wouldn't one just use full-parameter SFT, which also yields better performance?
    - If the objective is to address the difficulty of obtaining golden responses: The currently more popular "RL-Zero" approach also does not require golden responses. How does its performance compare to AWDPO? Of course, the authors might argue that AWDPO only requires a very small number of queries. I acknowledge this is an advantage, but it doesn't seem to be a significant one, as queries are relatively easy to obtain.

**Questions:**

See Strengths and  Weaknesses

**Details Of Ethics Concerns:**

nan

---

> ### Author Response · Authors · 2025-11-20
>
> **Official Comment by Authors**
>
> We thank the reviewer for the positive assessment of our method's elegance and performance, and for the insightful observation about connections to Policy Gradient methods.
>
> **Connection to Policy Gradient**
>
> We agree. AWDPO can indeed be viewed through a Policy Gradient lens. Our advantage-weighted loss shares structure with the REINFORCE algorithm when A(x) serves as the return.
>
> The key differences are:
> - Preference-based learning: We optimize relative probabilities between two outputs rather than absolute probabilities
> - No value estimation variance: Our advantage is deterministic (computed from fixed rewards), eliminating the high-variance gradient estimates typical in RL
> - Implicit baseline: The few-shot output serves as a natural baseline, making the advantage well-calibrated
>
> **W1.**
>
> We expanded our experimental scope to Llama 3.2 3B to showcase our method's applicability beyond the Qwen series in Section 6.
>
> **Multiple model architectures:**
>
> We now include Llama 3.2 3B results (Section 6):
>
> | Method | pass@1 | CoT Examples |
> |----|----|----|
> | DPO | 0.00% | 6,921 |
> | Filtered-DPO | 2.11% | 4 |
> | 4-shot | 0.07% | 4 |
> | **AWDPO** | **28.81%** | **4** |
> | SFT | 54.63% | 7,473 |
> | Answer-SFT | 8.78% | 4 |
>
> **Key findings:**
> 1. AWDPO bootstraps reasoning from severely weak teachers. Starting from 0.07% few-shot baseline, AWDPO improved Llama 3.2 3B to 28.81% on GSM8K (400× improvement), demonstrating robustness when latent reasoning is minimal.
> 2. Consistent mechanism across architectures. Llama 3.2 3B replicates the Qwen pattern: AWDPO outperforms Filtered-DPO (2.11%), standard DPO (0%), 4-shot prompting (0.07%), and Answer-SFT (8.78%) using only 4 CoT exemplars.
> 3. Architecture-dependent performance ceiling. Recovery versus SFT (54.63%) was 51%, lower than Qwen's 90-95%, indicating final performance is bounded by base model capacity. The AWDPO approach shows consistent relative gains across architectures. In sum, AWDPO successfully bootstraps reasoning even from severely degraded teachers.
>
> In addition, we add another benchmark for all models, which consists of supervised fine-tuning each model on 7,473 examples that consist of only a question and the answer without any chain-of-thought, and the same 4 chain-of-thought examples that we used in our AWDPO training, for a total of 7,477 training observations. This was meant to demonstrate how AWDPO compares to supervised fine-tuning with equal data budgets. A subset of the results on GSM8K can be found below:
>
> | Model | Answer-SFT | AWDPO |
> |----|----|----|
> | Qwen 2.5 0.5B | 5.35% | 38.83% |
> | Qwen 2.5 1.5B | 10.24% | 64.45% |
> | Qwen 2.5 3B | 14.40% | 77.63% |
>
> This demonstrates that with identical data budgets, AWDPO outperforms SFT by a large margin. Full results can be seen in Table 1 and Table 3.
>
> **Multiple datasets:** Models trained only on GSM8K generalize to SVAMP, ASDiv, and MATH-500 (Table 2), recovering 75-95% of SFT performance. We also evaluate on BBH non-math reasoning tasks (Appendix A, Tables 5-7), showing promising but mixed results that warrant future investigation.
>
> **W2a.**
>
> AWDPO uses LoRA (rank 64) which trains only ~0.3-1% of parameters compared to full fine-tuning. Based on our training runs, AWDPO with LoRA fits comfortably on consumer GPUs (24GB) while full SFT of 3B models requires high-end hardware. For example, full SFT of Qwen 2.5 3B took roughly 6 hours on a single A100 80GB, while AWDPO required only 4 hours on the same hardware with substantially lower memory requirements.
>
> **W2b.**
>
> AWDPO is designed for a different use case than full SFT or RL-Zero-style reinforcement learning. Full CoT-based SFT provides the best absolute accuracy, but it requires thousands of high-quality rationales and full-parameter or high-rank LoRA training.
>
> In contrast, AWDPO targets the low-supervision, low-compute regime where only a handful of exemplars are available and full-parameter training is infeasible.
>
> RL-Zero and related rollout-based RL methods such as DeepSeek-R1 assume mid- or large-scale models that already produce meaningful rollouts and therefore have usable reward gradients. They also require reward models, programmatic scoring, or large teacher models, and rely on extensive sampling infrastructure.
>
> These assumptions break down for small base models (0.5B-3B), which typically score 0-1% zero-shot due to a lack of instruction tuning and cannot generate viable trajectories for RL to learn from. AWDPO succeeds precisely in this setting: it improves small models by aligning their zero-shot behavior with their own few-shot reasoning, with minimal generation requirements, no external reward models, and preference-based learning instead of iterative RL training loops. Thus, AWDPO is not positioned as a substitute for large-scale RL pipelines but as an alternative when the base model is too weak for RL, when CoT data is scarce, or when compute budgets restrict training to lightweight PEFT methods.

---

### Official Review · Reviewer_YpX9 · 2025-11-01

**Soundness:** 2
**Presentation:** 3
**Contribution:** 2
**Rating:** 4
**Confidence:** 5

**Summary:**

This paper proposes Advantage-Weighted Direct Preference Optimization (AWDPO), a data-efficient method for transferring few-shot chain-of-thought reasoning into the zero-shot behavior of small language models (0.5B–3B) without requiring large teacher models or reinforcement learning. The key idea is to treat the model’s own few-shot outputs as a pseudo-teacher and compare them against its zero-shot outputs, weighting this preference by the advantage (difference in correctness-based reward), thereby enabling learning from both successful and unsuccessful reasoning attempts. Experimental results show that AWDPO achieves 39%–78% accuracy on GSM8K for 0.5B–3B Qwen-2.5 base models, recovering ~90% of the performance of a fully supervised fine-tune while using only four chain-of-thought exemplars. a 1,750× reduction in CoT data. The method generalizes to SVAMP, ASDiv, and MATH-500, where AWDPO recovers up to 90% of supervised CoT performance.

**Strengths:**

The paper focuses on improving zero-shot reasoning in small LLMs, which is a timely and practically important problem. The proposed advantage-weighted self-distillation formulation is conceptually clean: compare few-shot vs. zero-shot responses from the same model and update toward the one with higher reward. The proposed framework does not need the additional teacher model which significantly improves its efficiency. The paper demonstrates that low-rank adaptation (LoRA) inherently constrains policy drift, and a dynamic supervised anchor on correct few-shot examples further stabilizes training. With this, the additional regularization components such as KL divergence penalties, trust region methods or additional models etc are not needed. This makes the whole training pipeline much more lightweight.

**Weaknesses:**

The evaluation is very limited (only to GSM8K), so more comprehensive evaluations are definitely needed to validate the proposed methods.

AWDPO performance is still sensitive to the selection and phrasing of the few-shot prompts used to seed the pseudo-teacher. The paper would benefit from discussing guidance or robustness strategies for exemplar choice.

**Questions:**

How can we handle scenarios where the few-shot prompts are not representative? What's the underlying guidance for preparing the calibration datasets for self-distillation?

---

> ### Author Response · Authors · 2025-11-19
>
> We thank you for recognizing the timeliness and conceptual clarity of our work, and for highlighting the efficiency benefits of our framework.
>
> **W1. Limited Evaluation Scope**
>
> We thank you for your question and clarify that while we only train on GSM8K, the evaluation spans multiple datasets and benchmarks:
>
> **Cross-dataset generalization (Table 2):**
>
> | Dataset | 0.5B | 1.5B | 3B |
> |---------|------|------|-----|
> | GSM8K | 38.8% | 64.5% | 77.6% |
> | SVAMP | 40.8% | 69.1% | 79.1% |
> | ASDiv | 70.3% | 86.2% | 86.6% |
> | MATH-500 | 11.2% | 26.8% | 32.4% |
>
> Models trained only on GSM8K generalize to held-out datasets, recovering 75-90% of SFT performance without any additional training. We train only on GSM8K in order to measure the generalization ability of our method as it compares to SFT.
>
> **Non-math reasoning (BBH, Tables 5, 6, and 7):** We evaluate on Boolean Reasoning, Causal Judgment, and Logical Deduction across all model scales, demonstrating both strengths (competitive at 3B scale) and limitations (weaker at small scales) of the approach. A full hyperparameter sweep and analysis can be found in Tables 5, 6, and 7 in the Appendix.
>
> **Cross-architecture validation:** We now include Llama 3.2 3B results (Section 6), the new Table 3 shows cross-architecture validation:
>
> | Method | pass@1 | CoT Examples |
> |--------|--------|--------------|
> | DPO | 0.00% | 6,921 |
> | Filtered-DPO | 2.11% | 4 |
> | 4-shot | 0.07% | 4 |
> | **AWDPO** | **28.81%** | **4** |
> | SFT | 54.63% | 7,473 |
> | Answer-SFT | 8.78% | 4 |
>
> **Key findings:**
> 1. AWDPO bootstraps reasoning from severely weak teachers. Starting from 0.07% few-shot baseline, AWDPO improved Llama 3.2 3B to 28.81% on GSM8K (400× improvement), demonstrating robustness when latent reasoning is minimal.
> 2. Consistent mechanism across architectures. Llama 3.2 3B replicates the Qwen pattern: AWDPO outperforms Filtered-DPO (2.11%), standard DPO (0%), 4-shot prompting (0.07%), and Answer-SFT (8.78%) using only 4 CoT exemplars.
> 3. Architecture-dependent performance ceiling. Recovery versus SFT (54.63%) was 51%, lower than Qwen's 90-95%, indicating final performance is bounded by base model capacity. The AWDPO approach shows consistent relative gains across architectures. In sum, AWDPO successfully bootstraps reasoning even from severely degraded teachers.
>
> **W2 & Q1. Prompt Sensitivity and Non-Representative Exemplars**
>
> This is an important practical concern. Our approach includes several robustness mechanisms:
>
> 1. **Random sampling (k ∈ {2,3,4}):** We uniformly sample k exemplars from our fixed set of 4 for each training instance. This prevents overfitting to any single prompt configuration and implicitly tests robustness to exemplar subset selection.
>
> 2. **Advantage weighting filters poor prompts:** When exemplars are non-representative and produce low-quality few-shot outputs, A(x) will be small or negative, down-weighting their contribution to learning. The model learns primarily from instances where few-shot prompting actually helps.
>
> 3. **MLE anchor on correct examples:** Even if most few-shot outputs are poor, the MLE term ensures the model learns from the subset of correct demonstrations, preventing complete collapse of the few-shot teacher distribution.
>
> 4. To isolate the effect of self-distillation, we intentionally fixed a minimal exemplar set and avoided hand-curating prompts for maximal performance. This ensures the study reflects AWDPO's inherent robustness rather than prompt engineering.

---

### Official Review · Reviewer_Jfrx · 2025-11-01

**Soundness:** 3
**Presentation:** 3
**Contribution:** 2
**Rating:** 4
**Confidence:** 3

**Summary:**

This paper addresses the **lack of mathematical reasoning ability in small language models (0.5B–3B parameters)**, which typically perform poorly on reasoning benchmarks like GSM8K in zero-shot settings. The authors propose **Advantage-Weighted Direct Preference Optimization (AWDPO)** — a lightweight, single-pass self-distillation method that aligns a model’s zero-shot behavior with its own few-shot reasoning outputs. AWDPO computes a **preference loss weighted by the advantage (performance gap)** between few-shot and zero-shot responses, and combines it with a dynamically scaled **maximum likelihood (MLE) anchor** on correct examples. Experiments show AWDPO improves Qwen-2.5 models’ GSM8K accuracy from 0% to up to 77%, recovering over 90% of supervised fine-tuning performance with 1/1750th of the CoT data.

**Strengths:**

### **1. Data Efficiency**

AWDPO achieves performance gains using only **four chain-of-thought exemplars and answer-only supervision**, representing a **1,750× reduction in labeled CoT data** compared to fully supervised fine-tuning.

### **2. Clear Empirical Validation**

The authors provide **empirical validation** for some of the core claims of the paper, namely advantage weighting and dynamic loss-balancing through ablation studies.

**Weaknesses:**

### **1. Reliance on the Model’s Own Few-Shot Quality**

AWDPO assumes that the model’s few-shot responses are good enough to act as a “pseudo-teacher.” If the base model’s few-shot reasoning is poor, the entire self-distillation loop may propagate low-quality reasoning. The paper doesn’t explore how AWDPO behaves when the few-shot teacher is unreliable — e.g., for domains or smaller models where even few-shot reasoning fails.

### **2. Oversimplified Loss-Balancing Mechanism**

The online loss-balancing rule may be too heuristic and coarse-grained to ensure true gradient-scale equilibrium. It lacks theoretical justification or empirical exploration across task types, and may cause instability or suboptimal weighting when the two losses evolve at different rates (e.g. in a new reasoning task). The authors acknowledge dynamic scaling helps avoid manual tuning, but a more principled or adaptive method (e.g., gradient norm balancing) would strengthen the claim.

### **3. Inadequate Non-math Reasoning Results**

Only evaluation on non-math scenario can be found in Appendix. Even then, all baselines are missing for that setup.

**Questions:**

N/A

---

> ### Author Response · Authors · 2025-11-19
>
> We thank you for the constructive feedback and for recognizing our data efficiency gains and empirical validation.
>
> **W1. Reliance on Model's Few-Shot Quality**
>
> This is an excellent point. We now directly test AWDPO's robustness when the few-shot teacher is weak by examining the Llama 3.2 3B results (Table 3, Section 6):
>
> | Method | pass@1 | CoT Examples |
> |-------|-------|-------|
> | DPO | 0.00% | 6,921 |
> | Filtered-DPO | 2.11% | 4 |
> | 4-shot | 0.07% | 4 |
> | **AWDPO** | **28.81%** | **4** |
> | SFT | 54.63% | 7,473 |
> | Answer-SFT | 8.78% | 4 |
>
> The Llama results demonstrate that AWDPO achieves approximately ~400x relative improvement and validates the generalizability of our approach beyond the Qwen family.
>
> **Key findings:**
> 1. We observe that AWDPO successfully bootstraps reasoning even from severely degraded teachers. Starting from a few-shot baseline of less than 1%, AWDPO was able to improve Llama 3.2 3B to 28.81% on GSM8K, a more than 400x performance improvement.
> 2. Llama 3.2 3B shows a performance pattern similar to the Qwen models, with AWDPO outperforming Filtered-DPO, standard DPO, 4-shot prompting the base model, and fine-tuning on 7,473 answer-only observations with only 4 fixed chain-of-thought examples.
> 3. We also observe that when compared to SFT, Llama 3.2 3B has a more substantial performance gap than any of the Qwen series of models. This implies that although it is capable of bootstrapping reasoning out of small models, the final performance is bounded by the base model's capabilities.
> 4. Despite lower absolute performance compared to SFT, the pattern holds: AWDPO > Answer-SFT > Filtered-DPO > few-shot prompting, validating the method's core mechanism.
>
> The full table and analysis can be found in Section 6 and Table 3.
>
> **W2. Loss-Balancing Mechanism**
>
> We provide both theoretical justification and empirical validation.
>
> **Theoretical justification:** Our dynamic scaling λ_dyn = L_AWDPO / (L_MLE + ε) ensures that the two loss components contribute roughly equally to the total gradient magnitude. At equilibrium, ∇_θ L_total ≈ ∇_θ L_AWDPO + λ_dyn ∇_θ L_MLE, meaning both terms exert comparable gradient updates on the parameters. This prevents either objective from dominating and causing collapse or underfitting. While we agree that gradient norm balancing (scaling each loss to equalize ||∇_θ L_AWDPO|| and ||∇_θ L_MLE||) is more principled, it has practical drawbacks:
> - Requires computing full gradients before the update step (2× backward passes per iteration)
> - Can be unstable when gradients vary dramatically across mini-batches
> - In contrast, our loss-ratio scaling serves as a cheap proxy that correlates well with gradient magnitudes in practice
>
> Dynamic balancing consistently outperforms fixed weights and requires no hyperparameter tuning across model scales.
>
> **Empirical robustness:** Our method works consistently across three model scales (0.5B-3B) and multiple LoRA configurations without manual tuning of λ. Training curves (Figure 3) show stable convergence without oscillation or collapse.
>
> We acknowledge that gradient norm balancing would be more principled for future work. However, our simpler heuristic proves sufficient for the current scope (mathematical reasoning with stable reward signals). For domains with rapidly evolving or noisy rewards, more sophisticated balancing may be necessary.
>
> **W3. Non-Math Reasoning Results**
>
> We agree this deserved more prominence. We've expanded the BBH results in the appendix to demonstrate performance across hyperparameter configurations and at different pass@ levels (Section A.1) with a comparison to supervised fine-tuning, mirroring our analysis of out of sample math benchmarks (ASDiv, MATH 500, and SVAMP).
>
> AWDPO shows mixed results on non-math reasoning: it underperforms SFT at small scales but closes the gap at 3B, even exceeding SFT on Logical Deduction. Due to space constraints, we include the complete results in Tables 5, 6, and 7.
>
>
> | Method | Config | Boolean p@1 | Boolean p@5 | Causal p@1 | Causal p@5 | Logical p@1 | Logical p@5 | KL (nats) |
> |--------|--------|-------------|-------------|------------|------------|-------------|-------------|-----------|
> | SFT | SFT | 0.759 | 0.976 | 0.558 | 0.875 | 0.043 | 0.165 | --- |
> | AWDPO | r=64, α=64 | 0.635 | 0.943 | 0.320 | 0.724 | 0.064 | 0.250 | 0.007 |
> | AWDPO | r=64, α=128 | **0.727** | **0.958** | 0.428 | 0.821 | **0.083** | **0.303** | 0.021 |
> | AWDPO | r=128, α=128 | 0.668 | 0.946 | **0.492** | **0.882** | 0.053 | 0.212 | 0.017 |
> | AWDPO | r=64, α=256 | 0.695 | 0.947 | 0.404 | 0.784 | 0.036 | 0.139 | 0.232 |
>
> **Table:** Pass@1 and Pass@5 performance of AWDPO vs supervised fine-tuning (SFT) on Big Bench Hard subsets for Qwen 2.5 3B. AWDPO shows variable performance across configurations. KL divergence from base model is provided for each AWDPO configuration.
>
> We thank you for your comments that helped strengthen the paper.

---

### Official Review · Reviewer_FXUL · 2025-11-05

**Soundness:** 3
**Presentation:** 2
**Contribution:** 3
**Rating:** 4
**Confidence:** 3

**Summary:**

This paper introduces a fine-tuning technique called advantage-weighted direct preference optimization (AWDPO) that enables reasoning abilities (specifically GSM8K-style math reasoning) in small language models (Qwen-2.5 0.5-3B) using a much smaller dataset than techniques like SFT, while achieving similar performance.

**Strengths:**

* Building more data-efficient instruction-tuning methods for smaller language models is an important area of research.
* The paper contains many interesting experiments and analyses.
* The method seems theoretically grounded.

**Weaknesses:**

* The description of some experimental settings made it very difficult to follow the exact experimental setup and I am not 100% sure everything about the setup is sound based on the existing description. For example, the paper talks multiple times about QA pairs that were used as part of the fine-tuning data for AWDPO but it is never discussed what these examples are. Also, lines 250ff say "To increase prompt diversity for our methods, we randomly sample k ∈ { 2, 3, 4 } few-shot exemplars for each training instance." --> does this mean that more than 4 few-shot examples were used in total for AWDPO? If so, this would considerably weaken the data efficiency argument.
* Similarly, I think there should also be a comparison to SFT with the 4 chain-of-thought examples and the QA pairs for a fair comparison. Does AWDPO work better than that method?
* There is no discussion of computational efficiency. How does this method compare to SFT or PEFT techniques like LoRA?
* It would be good to know whether the method also works for other models.
* The setup of requiring several chain-of-thought examples and 7k QA pairs may not work very well for low-resource languages where such QA pairs may not be available. For higher-resource languages, on the other hand, it seems like it would be fairly easy to get CoT traces for existing data sets, so it is not entirely clear when this method would be useful in practice, which may limit impact.

**Questions:**

See questions regarding training details above. I'd be willing to raise my score a bit if the authors shared more details about the experimental setup in the author response (and they are sound).

---

> ### Author Response · Authors · 2025-11-19
>
> We thank you for recognizing the importance and theoretical grounding of our work, and for constructive feedback that significantly improved the paper's clarity.
>
> **W1. Experimental Setup Clarity**
>
> We apologize for the confusion and have revised Section 5 to clarify the exact training setup below:
>
> **Data:**
> - **4 CoT exemplars:** These are chain-of-thought solutions used only to prompt the model during data generation. They never enter training directly and are fixed (never more than 4).
> - **7,473 answer-only QA pairs:** GSM8K questions with answers (no reasoning traces). Provide questions (x) for generating few-shot (y+) and zero-shot (y−) responses.
>
> **Procedure:**
> 1. For each ~7k question, prompt base model with k randomly-selected exemplars from the fixed set of 4 COT exemplars → generate y+
> 2. Prompt same model zero-shot → generate y−
> 3. Compute advantage A(x) = R(x,y+) − R(x,y−)
> 4. Train via AWDPO loss on (y+, y−) pairs weighted by A(x)
>
> The k∈{2,3,4} sampling increases prompt diversity without using additional exemplars.
>
> **W2. Missing Baseline: SFT with 4 CoT + 7k QA**
>
> We trained this baseline:
>
> | Method | Data Used | GSM8K (3B) |
> |--------|-----------|------------|
> | Answer-SFT (4 CoT + 7k QA) | 4 CoT + 7k answer-only | 14.4% |
> | **AWDPO** | 4 CoT exemplars + 7k QA | **77.6%** |
> | SFT (Full) | 7,473 CoT traces | 81.8% |
>
> AWDPO's advantage-weighted learning extracts substantially more reasoning signal than standard supervised training from identical data. Full results for all Qwen 2.5 and Llama 3.2 model sizes appear in Tables 1 and 3.
>
> **W3. Computational Efficiency**
>
> AWDPO uses LoRA (rank 64) which trains only ~0.3-1% of parameters compared to full fine-tuning. Based on our training runs, AWDPO with LoRA fits comfortably on consumer GPUs (24GB) while full SFT of 3B models requires high-end hardware. For example, full SFT of Qwen 2.5 3B took roughly 6 hours on a single A100 80GB, while AWDPO required only 4 hours on the same hardware with substantially lower memory requirements.
>
> **W4. Other Models - Llama 3.2 3B**
>
> New Table 3 in Section 6 shows cross-architecture validation:
>
> | Method | pass@1 | CoT Examples |
> |--------|--------|--------------|
> | DPO | 0.00% | 6,921 |
> | Filtered-DPO | 2.11% | 4 |
> | 4-shot | 0.07% | 4 |
> | **AWDPO** | **28.81%** | **4** |
> | SFT | 54.63% | 7,473 |
> | Answer-SFT | 8.78% | 4 |
>
> **Key findings:**
> 1. AWDPO bootstraps reasoning from severely weak teachers. Starting from 0.07% few-shot baseline, AWDPO improved Llama 3.2 3B to 28.81% on GSM8K (400× improvement), demonstrating robustness when latent reasoning is minimal.
> 2. Consistent mechanism across architectures. Llama 3.2 3B replicates the Qwen pattern: AWDPO outperforms Filtered-DPO (2.11%), standard DPO (0%), 4-shot prompting (0.07%), and Answer-SFT (8.78%) using only 4 CoT exemplars.
> 3. Architecture-dependent performance ceiling. Recovery versus SFT (54.63%) was 51%, lower than Qwen's 90-95%, indicating final performance is bounded by base model capacity. The AWDPO approach shows consistent relative gains across architectures. In sum, AWDPO successfully bootstraps reasoning even from severely degraded teachers.
>
> **W5. Practical Applicability and Low-Resource Languages**
>
> AWDPO requires 7k answer-only QA pairs. When questions are unavailable (true low-resource scenarios), neither AWDPO nor SFT works. However, requirements differ:
> - **SFT:** 7k questions + 7k answers + 7k CoT traces
> - **AWDPO:** 7k questions + 7k answers + 4 CoT traces
>
> When questions exist (translated problems, exam datasets, technical domains) but expert CoT annotation is the bottleneck, AWDPO's 1/1,750× reduction in CoT requirements provides value. Generating quality CoT data requires either high-end hardware + engineering effort for filtering, or substantial API costs. AWDPO requires only 4 exemplars and lower-end hardware.
>
> Additionally, there still exist substantial barriers to generating high quality CoT data, even for existing datasets. Generating CoT data requires either renting/purchasing high-end hardware and engineering effort for quality filtering, or substantial API costs from closed-source models. AWDPO has the practical benefit of only requiring 4 exemplars and lower-end hardware as opposed to substantial infrastructure and data acquisition costs.
>
> The clarified setup, new Llama results, requested baseline comparison, and efficiency analysis provide strong evidence that AWDPO is a practical, generalizable method for data-efficient reasoning transfer. We thank you for your comments that helped strengthen the paper. We hope you find the revised version satisfactory.

---

### Author Response · Authors · 2025-11-20

****

**Summary of Key Changes for Area Chair**

We addressed all reviewer comments. Specifically, we generalized our approach to Llama 3.2 3B and achieved 400x improvement in model performance; following the reviewers' comments, we added a new baseline benchmark Answer-SFT and showed our approach performed 5-7x better; expanded on the big bench hard results. We also note that reviewer R-FXUL offered to raise the score if setup clarified, which we have done. We respond to the scope concern of R-XGYQ by adding a new model and a new benchmark. For R-Jfrx's weak-teacher question we generalized to Llama 3.2 3B and show a 400x gain. Finally, we clarified R-YpX9 comment that we only used GSM8K - Table 2 shows 4-dataset generalization; which is now further strengthened. We addressed all the reviewers' comments and have completed both cross-architecture and cross-dataset validation. We hope the revisions meet with your approval.

****

**Comment to All Reviewers**

We sincerely thank the reviewers for their thoughtful and constructive feedback. We are encouraged that reviewers found our method data-efficient (all reviewers), theoretically grounded (R-FXUL), conceptually clean (R-YpX9, R-XGYQ), and well-validated empirically (R-Jfrx). We have revised the paper to address the concerns raised, expanding the experimental scope and clarifying critical details about the training setup.

**Key Updates**

**1. Expanded Model Scope (R-FXUL, R-Jfrx, R-YpX9, R-XGYQ):** We address the primary concern about limited model coverage by training and evaluating AWDPO on Llama 3.2 3B. On GSM8K, Llama 3.2 3B improved zero-shot accuracy with AWDPO by ~400x relative improvement. This demonstrates that AWDPO's effectiveness generalizes beyond the Qwen family to other base model architectures. The Llama results, along with updated tables and analysis, are presented in Section 6.

**2. Multi-Dataset Training and Evaluation:** We clarify that while our training experiment used only GSM8K for supervised fine-tuning and AWDPO (for controlled comparison with baselines), AWDPO demonstrates strong zero-shot generalization to multiple datasets without retraining. On SVAMP, ASDiv, and MATH-500 AWDPO recovers 90-95% of fully-supervised performance on these held-out benchmarks (Table 2). This demonstrates that the reasoning patterns internalized by AWDPO transfer broadly.

**3. Training Setup Clarification (R-FXUL):** We describe the experimental setup in Section 5. To clarify: AWDPO uses a fixed set of 4 chain-of-thought observations that are only used as few-shot examples at training time, never as part of the training data itself. The notation k ∈ {2,3,4} refers to sampling a random subset of between 2 and 4 exemplars per training instance to increase prompt diversity, not additional exemplars. This maintains our data efficiency claim while ensuring robust learning. The QA pairs refer to the GSM8K training questions (approximately 7k) for which we generate both few-shot and zero-shot completions from the base model to create preference pairs. These are not additional labeled data, they are simply the questions used to collect model outputs for training. The ground-truth answers (already available in GSM8K) are used only to compute the advantage via our reward function, not as supervised targets for generation.

---

### Meta-Review · Area_Chair_nKzz · 2025-12-18

**Summary:**

Following the rebuttal, several significant concerns remain unresolved:

First, as highlighted by Reviewers Jfrx, YpX9, and XGYQ, the experimental scope remains narrow. The evaluation relies on only three models within the same family (Qwen 2.5) trained on a single dataset (GSM8K). While the authors incorporated some preliminary results—specifically including the Llama 3.2 8B model and mixed results from additional logical reasoning tasks—the empirical evidence is not convincing enough. To substantiate the paper's claims, a more comprehensive evaluation involving a wider variety of model architectures and diverse tasks, such as coding and general QA, is required.

Second, the critique raised by Reviewers FXUL and XGYQ regarding the limited applicability of the proposed method is a critical issue. Although the authors argue that their approach is specifically designed for low-supervision and low-compute regimes, this narrow focus appears to constrain the method's utility and impact. Consequently, it remains difficult to justify the practical value of the work.

**Reviewer Concerns:**

The rebuttal has partially addressed the following concerns:
- Unclear writing of experimental setup. [FXUL]
- Missing comparison with SFT. [FXUL]
- Missing discussion on computational efficiency. [FXUL]
- Unclear generalization to other models. [FXUL]
- Oversimplified loss-balancing mechanism. [Jfrx]

The remaining concerns are still outstanding:
- Limited practical usage due to the setup. [FXUL, XGYQ]
- Reliance on the model’s own few-shot quality. [Jfrx, YpX9]
- Limited experimental scope, e.g., only 3 models within the same family on math reasoning. [Jfrx, YpX9, XGYQ]
- What's the underlying guidance for preparing the calibration datasets for self-distillation? [YpX9]

**Reviewer Scores:**

Given the full discussion, the Reviewer FXUL may consider changing the score due to the clarification on the detailed description of experimental setup.
However, for the remaining reviewers, it is unlikely that the rebuttal would have altered their final assessments even with full participation in the discussion. The fundamental concerns regarding the limited experimental scope and constrained applicability of the method remain largely unaddressed.

---

### Decision · Program_Chairs · 2026-01-26

Reject